# A biallelic *SNIP1* Amish founder variant causes a recognizable neurodevelopmental disorder

Zineb Ammous[1]*, Lettie E. Rawlins[2,3], Hannah Jones[2], Joseph S. Leslie[2], Olivia Wenger[4], Ethan Scott[4], Jim Deline[5], Tom Herr[5], Rebecca Evans[1], Angela Scheid[1], Joanna Kennedy[2], Barry A. Chioza[2], Ryan M. Ames[6], Harold E. Cross[7], Erik G. Puffenberger[8], Lorna Harries[2], Emma L. Baple[2,3]*, Andrew H. Crosby[2]*

**1** The Community Health Clinic, Topeka, Indiana, United States of America, **2** Medical Research, RILD Wellcome Wolfson Centre, University of Exeter Medical School, Royal Devon & Exeter NHS Foundation Trust, Exeter, United Kingdom, **3** Peninsula Clinical Genetics Service, Royal Devon & Exeter Hospital (Heavitree), Exeter, United Kingdom, **4** New Leaf Center, Clinic for Special Children, Mount Eaton, Ohio, United States of America, **5** Center for Special Children, La Farge Medical Center, La Farge, Wisconsin, United States of America, **6** Biosciences, Geoffrey Pope Building, University of Exeter, Exeter, United Kingdom, **7** Department of Ophthalmology, University of Arizona College of Medicine, Tucson, Arizona, United States of America, **8** Clinic for Special Children, Strasburg, Pennsylvania, United States of America

* zammous@indianachc.org (ZA); E.Baple@exeter.ac.uk (ELB); A.H.Crosby@exeter.ac.uk (AHC)

**Data Availability Statement:** The data used for the analyses described in this manuscript (S2 and S3 Figs) were obtained from: https://gtexportal.org/home/multiGeneQueryPage the GTEx Portal on 07/

## Abstract

SNIP1 (Smad nuclear interacting protein 1) is a widely expressed transcriptional suppressor of the TGF-β signal-transduction pathway which plays a key role in human spliceosome function. Here, we describe extensive genetic studies and clinical findings of a complex inherited neurodevelopmental disorder in 35 individuals associated with a *SNIP1* NM_024700.4:c.1097A>G, p.(Glu366Gly) variant, present at high frequency in the Amish community. The cardinal clinical features of the condition include hypotonia, global developmental delay, intellectual disability, seizures, and a characteristic craniofacial appearance. Our gene transcript studies in affected individuals define altered gene expression profiles of a number of molecules with well-defined neurodevelopmental and neuropathological roles, potentially explaining clinical outcomes. Together these data confirm this *SNIP1* gene variant as a cause of an autosomal recessive complex neurodevelopmental disorder and provide important insight into the molecular roles of SNIP1, which likely explain the cardinal clinical outcomes in affected individuals, defining potential therapeutic avenues for future research.

## Author summary

Neurodevelopmental disorders are a group of conditions that may be inherited in families, characterized by impairments of the growth, development and function of the brain. This may result in neuropsychiatric problems, impaired motor function, impaired learning, language and/or non-verbal communication. These conditions may be associated with epilepsy, characterised by recurrent abnormal electrical activity in the brain. Here we confirm a founder *SNIP1* gene variant as a cause of an autosomal recessive complex

01/2021. All relevant deidentified data are within the manuscript and its Supporting Information files.

**Funding:** This work was supported by MRC grant G1001931 (to ELB), MRC grant G1002279 (to AHC), MRC Proximity to Discover and Confidence in Concept grants (MC-PC-18047, MC_PC_15054, MC_PC_15047 to University of Exeter, ELB and AHC) https://mrc.ukri.org/funding/, and the Newlife Foundation for disabled children (AHC, ELB and LER) https://newlifecharity.co.uk/index.php. RMA is supported by a BBSRC/EPSRC Interface Innovation Fellowship (EP/S001352/1) https://www.ukri.org/opportunity/?filter_council%5B%5D=816. The funders had no role in study design, data collection and analysis, decision to publish, or preparation of the manuscript.

**Competing interests:** The authors have declared that no competing interests exist.

neurodevelopmental disorder in the Amish. We provide a detailed description of the clinical features of the condition alongside clinical management recommendations. Our genetic studies identify altered gene expression patterns in affected children associated with the *SNIP1* genetic alteration. This identified abnormalities in several proteins with important roles in brain development and function, potentially explaining the clinical features of the condition.

## Introduction

Smad proteins entail a family of structurally-related intracellular proteins that act as transcriptional regulators of the transforming growth factor-beta (TGF-β)/bone morphogenetic protein (BMP) pathway, crucially important for a variety of cellular processes including cell growth, differentiation and apoptosis [1]. Three categories of Smad are recognised: receptor-regulated (R-Smads) and common mediator (Co-Smads), which form trimeric functional units of two R-Smads with the common Co-Smad partner SMAD4 and inhibitory/anti (I-Smads), acting together to inhibit these functional units [2].

The *Smad nuclear interacting protein 1 (SNIP1)* gene encodes a widely expressed nuclear transcriptional regulator of several important molecular signalling pathways involved in embryogenesis. This evolutionarily conserved protein is made up of 396 amino acids, comprising a two-part nuclear localisation signal (NLS) and a forkhead-associated (FHA) domain (Fig 1H) [3,4]. The N-terminus NLS interacts with several Smads in the TGF-β signalling pathway [3], and inhibits the p65/RELA subunit of nuclear factor kappa-beta (NFkB; a primary transcription factor in proinflammatory signalling pathways) to act as a transcriptional repressor [4]. The SNIP1 N-terminal domain also acts to inhibit transcription via competitive binding of transcriptional co-activators CREBBP (CREB-binding protein) and EP300 (E1A-binding protein, 300 kDa) [3,4]. The C-terminal FHA domain of SNIP1 has been shown to enhance transcriptional activity via direct binding to the oncoprotein c-MYC (a key regulator of cell proliferation and growth-promoting pathways) [5]. Further molecular pathways that may be regulated by SNIP1 activity include those involved in carcinogenesis, including regulation of cyclin D1 expression [6] and ATR checkpoint kinase-dependent pathways [7], as well as heat shock elements through regulation of heat shock factor-mediated transcription of heat shock proteins [8]. SNIP1 also forms a component of the human activated spliceosome [9], as part of the retention and splicing (RES) complex, which plays a crucial role in splicing and retention of pre-mRNA [10]. Genetic variants leading to disruption of the TGF-β/BMP molecular signalling pathway have been implicated in the pathogenesis of a number of inherited disorders, in particular connective tissue disorders associated with craniofacial (e.g. cleft palate and craniosynostosis), cardiovascular (e.g. aortic aneurysm and syndromic aortopathies), and musculoskeletal (e.g. disorders of endochondral ossification, bone growth, remodelling and limb development) outcomes [11].

The Amish are descended from a relatively small group of Swiss-German individuals that emigrated from Europe to the USA in two main waves of migration in the 18th and 19th centuries. Puffenberger et al (2012) previously described three Old Order Amish individuals with symptomatic epilepsy and skull dysplasia and identified a homozygous NM_024700.4: c.1097A>G, p.(Glu366Gly) variant in *SNIP1* as the likely cause of the condition (autosomal recessive psychomotor retardation, epilepsy, and craniofacial dysmorphism; PMRED, MIM #614501) [12]. These three individuals displayed neonatal hypotonia, severe global developmental delay, dysmorphic features, a "lumpy" skull surface, poor feeding, intractable seizures, brain abnormalities on MRI, and other congenital defects. Here we present comprehensive

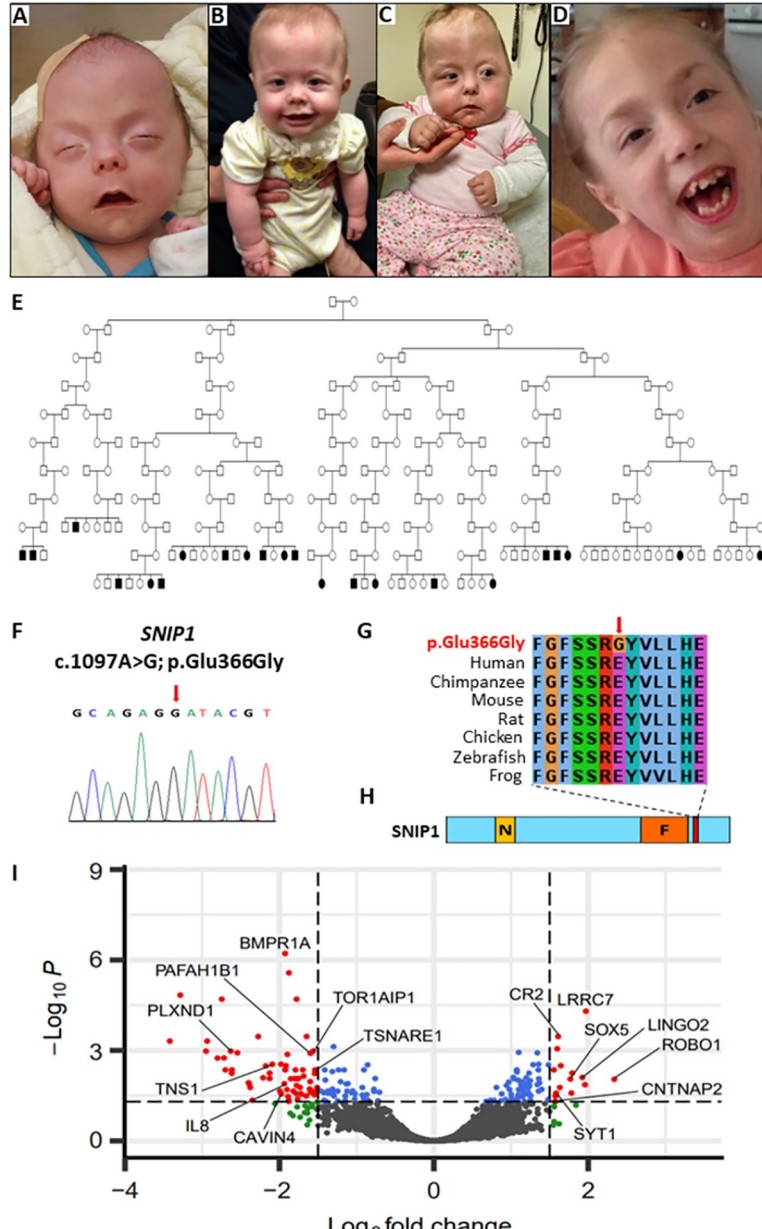

**Fig 1. Homozygous *SNIP1* c.1097A>G; p.(Glu366Gly) variant results in multisystem syndromic *SNIP1*-related disorder within the Amish community (A-D)** Clinical features of individuals affected by *SNIP1*-related disorder. **(A-C)** Clinical features of three affected infants comprising of **(A)** craniosynostosis, proptosis, **(A-C)** downslanting palpebral fissures, microretrognathia, cupids bow upper lip and small upturned bulbous nose **(B)** hypotonia, tracheostomy, and gastrostomy feeding tube **(C)** metopic ridge, asymmetrical skull shape. **(D)** Photograph of older child, additional phenotypic features illustrated include; micrognathia, wide-mouth and dental crowding. **(E)** Simplified pedigree of 12 interlinking Amish nuclear families illustrating 22 individuals (of 35 investigated, which could be linked into a single pedigree using available Amish ancestry databases) affected with *SNIP1*-related disorder and confirmed to be homozygous for the *SNIP1* NM_024700.3: c.1097A>G, p.(Glu366Gly) variant. **(F)** Sequence chromatogram of the c.1097A>G, p.(Glu366Gly) variant (red arrow). **(G)** Multi-species alignment showing conservation of the polypeptide region encompassing the p.(Glu366Gly) alteration (red arrow). **(H)** Schematic diagram of the SNIP1 protein identifying the N-terminus nuclear localisation signal (NLS) domain (yellow box), C-terminus forkhead-associated (FHA) domain (orange box) and location of the c.1097A>G, p.(Glu366Gly) variant (red box). **(I)** Visualisation of differential expression results showing genes grouped by log(fold change) (x-axis) and log (adjusted p-value) (y-axis). Dashed vertical lines show the log(fold change) cut-off used to identify differentially expressed genes showing the largest relative expression changes (green points). The dashed horizontal line shows the log(adjusted p-value) cut-off used to identify significantly differentially expressed genes (blue points). Genes that are

both significantly differentially expressed and show the largest change in expression are coloured red. Non-significant genes with low changes in expression are shown in grey. A selection of differentially expressed genes are labelled.

genetic, clinical and gene transcript functional outcomes stemming from the investigation of 35 Amish individuals with an autosomal recessive neurodevelopmental disorder, corroborating this *SNIP1* gene variant as a cause of this disease.

# Results

## Clinical features

35 of 51 identified individuals from 21 interrelated Old Order Amish families (19 males; 16 females, aged 1 month to 26 years) with molecularly confirmed *SNIP1*-related disorder were evaluated (see Fig 1, and Table 1 for clinical summary, includes new findings for two affected

**Table 1. Clinical features of 35 individuals with *SNIP1*-related disorder.**

|  | Mean age in years (range) | 6.9 (0.2–26) |
|---|---|---|
| **Neurological phenotype** |  | **(% affected)** |
|  | Global developmental delay (severe, non-verbal) | 100% |
|  | Hypotonia | 100% |
|  | Hyporeflexia | 100% |
|  | Seizures | 100% |
|  | Abnormal brain MRI (ventriculomegaly, white matter defects) | 50% |
|  | Behavioural problems (irritability, autistic features, ADHD) | 75% |
| **Cardiopulmonary features** |  |  |
|  | Upper airway abnormalities (laryngomalacia, apnoea, stridor) | 75% |
|  | Congenital heart defects (ASD, VSD, aortic coarctation) | 60% |
|  | Cardiomyopathy | 12% |
| **Visual and hearing phenotype** |  |  |
|  | Horizontal nystagmus and/or strabismus | 45% |
|  | Failed newborn hearing screen (conductive loss) | 21% |
| **Gastroenterological features** |  |  |
|  | Feeding difficulties | 100% |
|  | Small for gestational age | 54% |
|  | Pulmonary aspiration with/without gastrostomy tube | 46% |
| **Endocrine phenotype** |  |  |
|  | Hypothyroidism | 25% |
|  | Hypoglycaemia | 21% |
| **Dysmorphic features** |  |  |
|  | Abnormal skull shape (irregular surface, craniosynostosis) | 100% |
|  | High arched palate | 100% |
|  | Wide mouth with cupids bow upper lip | 100% |
|  | Micrognathia with/without Pierre Robin sequence | 30% |
|  | Short hands with tapered fingers | 54% |
|  | Spinal abnormalities (scoliosis, sacral dimple, tethered cord) | 21% |
|  | Hernias (umbilical, inguinal) | 21% |
|  | Congenital talipes equinovarus | 13% |

Abbreviations: MRI; magnetic resonance imaging, ADHD; attention deficit hyperactivity disorder, ASD; atrial septal defect, VSD; ventricular septal defect.

individuals previously published by Puffenberger et al 2012 [12]). Six children died between ages 9 months and 11 years as a consequence of infection, sudden cardiopulmonary arrest, or accidental drowning. Substantial phenotypic variation was observed between affected individuals, including within sibships.

The cardinal clinical features of *SNIP1*-related disorder include: hypotonia, global developmental delay, intellectual disability, seizures, a characteristic facial appearance and skull abnormalities. Craniofacial features (Fig 1A–1D) include midface hypoplasia, a wide mouth with downturned corners and thin cupids bow upper lip, large tongue, high arched palate, microretrognathia (Pierre Robin sequence with or without cleft palate in three patients), malocclusion, small upturned bulbous nose, long palpebral fissures and proptosis (in individuals with craniosynostosis). Other dysmorphic features include brachydactyly, broad thumbs, single palmar crease, congenital talipes equinovarus, pes planus, toenail hypoplasia and pectus excavatum (Table 1). Skull morphological abnormalities were observed and ranged from normocephaly to severe multi-suture craniosynostosis with Cloverleaf appearance of the skull in five individuals, and included an irregular skull surface with variable calvarium thickness and bony dysplasia (Fig 1A–1D).

The neurological spectrum of the syndrome is variable and includes seizures, hypotonia with hypo/areflexia and pervasive behavioural abnormalities. Seizures are a cardinal feature of the disorder with all affected individuals developing epilepsy. Seizure types include focal and generalised intractable seizures (myoclonic, absence, tonic-clonic) of infantile or childhood onset. EEG was abnormal in all individuals assessed, with no characteristic pattern, findings included diffuse background slowing and frequent focal or multifocal discharges (see S1 Text for detailed serial EEG findings observed in eight affected individuals). In some infants, seizures initially presented as apnea or breath-holding spells and subclinical seizures were detected in six individuals. Seizures can be triggered or exacerbated by febrile illness, sleep deprivation and hypoglycemic episodes. There were no antiepileptic medications identified that consistently provide effective seizure control (defined as manageable frequency of seizures) and several individuals display multiple drug resistant epilepsy. Some individuals suffered severe developmental regression following status epilepticus, with two reported to have subsequent permanent hemiparesis. Individuals identified in late childhood/adolescence are often diagnosed with a mixed tone spastic quadriplegic cerebral palsy, likely reflecting hypoxic injury after long-term untreated/unrecognised seizures. Irritability and sleep/wake dysregulation are seen in infancy and early childhood. Behavioural diagnoses include autism spectrum disorder and attention deficit disorder. Most affected individuals achieved independent ambulation (age range 3–10 years), and some communicate with signs, gestures and sounds. Neuroimaging was available for 18 individuals, all confirmed to have abnormalities, including hydrocephalus, ventriculomegaly, white matter abnormalities, thin corpus callosum, hypomyelination, irregular cortical ribbon, Chiari malformation, absence of the septum pellucidum, hypoplastic optic nerves and septo-optic dysplasia [12]. Cerebral arteriovenous malformation (AVM) was reported in a single individual. Other clinical features associated with *SNIP1*-related disorder include a variable spectrum of congenital cardiac defects (60% of individuals), including hypoplastic left heart syndrome, aortic stenosis and bicuspid aortic valve, aortic coarctation, aortic root dilatation, atrial septal defect (ASD), ventricular septal defect (VSD), patent ductus arteriosus (PDA), pulmonary artery stenosis and mitral valve regurgitation, 12% of affected individuals developed cardiomyopathy (left ventricular noncompaction). Affected individuals commonly displayed upper airway respiratory difficulties in the neonatal period, including laryngomalacia, pharyngomalacia and subglottic stenosis of variable severity, causing a weak cry, stridor and apnea, occasionally requiring tracheostomy and supplementary home oxygen.

Spinal abnormalities were seen in 23% including sacral dimple, tethered cord and scoliosis in older children. Other features include pectus excavatum and hypoplastic ribs in a single individual. Poor feeding in the neonatal period is universal and many require a gastrostomy tube to both aid nutrition and reduce risk of aspiration pneumonia. Gastrointestinal features reported include gastroesophageal reflux, paraoesophageal hernia (type III), intestinal malrotation, hepatomegaly and umbilical and inguinal hernias. Intrauterine growth retardation (IUGR) and failure to thrive was also common. Endocrine abnormalities including hypoglycemia, hypothyroidism, hypothermia and dyshormonogenesis were identified in a small proportion of patients. Ophthalmic features include strabismus, horizontal nystagmus and myopia. Auditory features include chronic otitis media and hearing loss, with 20% failing newborn hearing screening.

## Genetic studies

Our initial studies involved the investigation of four affected individuals (X:30, X:31, XI:6 and XI:7) from two nuclear families, on whom samples were originally available (Fig 1E). In order to map the likely chromosomal location of the causative gene a genome-wide screen was undertaken to identify putative autozygous genomic regions, assuming homozygosity for the same ancestral founder variant, which entails the most common cause of recessive monogenic disorders in the Amish. Inspection of resultant genotypes identified a single region of homozygosity greater than 1Mb common to all four affected individuals, a small (1.65Mb) genomic region on chromosome 1p34.3 delimited by SNP markers rs6667450 and rs10889902 (NC_000001.11:g.36,492,230–38,143,653; S1 Fig), highly likely to contain the disease locus, comprising 17 genes. To identify candidate genetic variants in this region WES data on two individuals (X:16 and X:42) identified only a single potentially pathogenic sequence variant (Chr1:g.37537842T>C; RefSeq NM_024700.4:c.1097A>G, p.(Glu366Gly) [hg38]) in exon four of the *SNIP1* gene, located in the chromosome 1p34.3 critical interval, previously identified by Puffenberger et al 2012 [12]. Dideoxy sequencing validated the presence of the variant (Fig 1F), which cosegregated as appropriate for an autosomal recessive condition. Subsequent targeted genetic investigation of additional affected individuals that were identified, confirmed homozygosity of the *SNIP1* variant in 35 individuals, with all parents being heterozygous for the variant and all unaffected siblings being heterozygous for the variant or wild type (Fig 1E). Multiple species alignment confirms this amino acid is highly conserved across species orthologs (Fig 1G) and is located close to the FHA domain at the SNIP1 C-terminus (Fig 1H). Inspection of our in-house Amish control database (>5000 individuals) determined that the *SNIP1* NM_024700.4:c.1097A>G, p.(Glu366Gly) variant is present at an allele frequency of between 0.5% (Pennsylvania Amish) and 1.4% (Ohio/Indiana/Wisconsin Amish) in this founder population, with no unaffected individuals being homozygous for this variant. The variant is present in gnomAD (v3.1) with an allele frequency of 0.001% with 11 heterozygotes (10 Amish) and no homozygous individuals listed, with multiple in silico tools predicting the variant to be pathogenic. The *SNIP1* variant was assessed according to ACMG variant classification criteria as 'pathogenic' (see S2 Text).

## Effect of *SNIP1* variant on gene expression and cellular pathways

In order to provide improved insight into *SNIP1* gene function, and outcomes of the NM_024700.4:c.1097A>G; p.(Glu366Gly) gene variant on gene expression, we performed whole transcriptome sequencing on six affected individuals from four distinct nuclear sibships, alongside sex-matched controls. We performed a power estimation using the characteristics of our RNA-Seq data that included sequencing depth, dispersion and log fold change to estimate

**Table 2. List of the 10 most differentially expressed (both up/downregulated) gene pathways.**

| Pathway name | Pathway identifier | #Entities found | #Entities total | Entities ratio | Entities pValue | Entities FDR | Genes |
|---|---|---|---|---|---|---|---|
| TGF-beta receptor signaling in EMT (epithelial to mesenchymal transition) | R-HSA-2173791 | 3 | 19 | 0.00129 | 0.002827 | 0.43317 | ***PARD3**; RPS27A; F11R* |
| Transcriptional regulation of pluripotent stem cells | R-HSA-452723 | 4 | 45 | 0.00306 | 0.004461 | 0.43317 | *NR5A1; **EPAS1**; **ATP9A*** |
| RUNX1 regulates transcription of genes involved in BCR signaling | R-HSA-8939245 | 2 | 7 | 0.00048 | 0.004820 | 0.43317 | ***BLK*** |
| CDC6 association with the ORC:origin complex | R-HSA-68689 | 2 | 11 | 0.00075 | 0.011458 | 0.43317 | *E2F2; E2F3* |
| Signaling by NOTCH3 | R-HSA-9012852 | 4 | 63 | 0.00428 | 0.014027 | 0.43317 | ***PLEKHG1**; PLXND1; RPS27A* |
| Pexophagy | R-HSA-9664873 | 2 | 13 | 0.00088 | 0.015702 | 0.43317 | ***EPAS1**; RPS27A* |
| NOTCH3 Intracellular Domain Regulates Transcription | R-HSA-9013508 | 3 | 36 | 0.00245 | 0.016072 | 0.43317 | ***PLEKHG1**; PLXND1* |
| Erythrocytes take up oxygen and release carbon dioxide | R-HSA-1247673 | 2 | 16 | 0.00109 | 0.023123 | 0.43317 | *SLC4A1; HBD* |
| Oncogene Induced Senescence | R-HSA-2559585 | 3 | 42 | 0.00285 | 0.023970 | 0.43317 | *E2F2; E2F3; RPS27A* |
| Defective binding of RB1 mutants to E2F1,(E2F2, E2F3) | R-HSA-9661069 | 2 | 17 | 0.00115 | 0.025859 | 0.43317 | *E2F2; E2F3* |

List of the top 10 human gene pathways that are most differentially expressed (both up/downregulated) from Reactome [13] overrepresentation analysis of significantly up/downregulated genes in six affected individuals compared with sex-matched controls and a *log(fold change)* >1.5 were used in gene set enrichment analysis of pathway and gene ontology. Pathways with an entities P value of <0.05 were considered significantly overrepresented. Genes in **bold** were upregulated, those in normal font were downregulated. FDR, false discovery rate is the probability corrected for multiple comparisons.

the probability of finding differentially expressed genes in our data. This power calculation indicated a >50% probability (power = 0.59) of finding significantly differentially expressed genes given our sample size and data characteristics, confirming that the number of samples and quality of the RNA-Seq data are sufficient to identify differentially expressed genes.

Differentially expressed genes with a false discovery rate (FDR) adjusted p-value of <0.05 identified a list of 75 significantly upregulated genes, and 109 significantly downregulated genes (Tables A and B in S1 Table). A log fold change of >1.5 was used as an additional cut off for both up and downregulated genes to select only those that show the largest relative changes in expression (Fig 1I). These genes were assessed for known clinical associations and involvement in conditions with overlap with the clinical features in affected individuals enrolled on this study (Tables A and B in S2 Table). Additionally Reactome pathway analysis [13] combining both up and downregulated genes identified a total of 30 significantly overrepresented pathways (entities p value <0.05) (Tables A-C in S3 Table); the most overrepresented pathway was the TGF-β receptor signalling in epithelial to mesenchyme pathway with differential expression of three genes (upregulation of PARD3 and downregulation of F11R and RPS27A) (Table 2). Tissue expression of genes up/downregulated in individuals with *SNIP1*-related disorder was assessed using GTex, Broad Institute 2021 (S2 and S3 Figs), which identified that several of the upregulated genes are highly expressed within the brain, although this was not apparent for downregulated genes.

## Discussion

Here, we describe our studies of a complex syndromic autosomal recessive neurodevelopmental disorder due to a biallelic *SNIP1* variant (NM_024700.4:c.1097A>G, p.(Glu366Gly)), which

we found to be common amongst the Old Order Amish. A total of fifty-one Old Order Amish individuals were identified with *SNIP1*-related disorder, among 27 sibships. Seventy-six percent (n = 39) are currently living (age range 6 months to 30 years), while 24% (n = 12) are deceased (age of death between 1 week and 11 years). Forty-nine individuals were confirmed to be homozygous for the *SNIP1* variant, two individuals were not genotyped, but displayed characteristic clinical features of the disorder and had an affected sibling confirmed to be homozygous for the *SNIP1* founder variant. In total thirty-five individuals with *SNIP1*-related disorder were comprehensively clinically evaluated and included in this study.

The condition involves a recognizable neurodevelopmental phenotype comprising of a characteristic craniofacial appearance, skull abnormalities, developmental delay, intellectual disability and variable additional congenital malformations, neurological, cardiovascular, respiratory and gastrointestinal features. Seizures, starting in infancy, are a universal feature and can include any combination of focal or generalised, myoclonic, absence or tonic-clonic forms. Our findings determine that clinical management strategies should be targeted at preventing life-threatening complications and optimising psychomotor development and are summarized in Table 3.

We recommend elective gastrostomy tube placement to support growth and limit pulmonary aspiration, as well as careful optimisation of anticonvulsant medications to treat apnea in infancy, maintain seizure control, and prevent status epilepticus and consequent

**Table 3. *SNIP1*-related disorder clinical advice and guidelines.**

| | |
|---|---|
| **Medical care:** | Children should be under the care of a general or community paediatrician to monitor their general health and development. Clinical management strategies should be targeted at preventing life-threatening complications and optimising psychomotor development. |
| **Seizures:** | Children should be under the care of specialist neurology services for careful optimisation of anticonvulsant medications to treat apnea in infancy, maintain seizure control, and prevent status epilepticus and consequent developmental regression. Early EEG should be carried out in infants with suspected apneic episodes. Neuroimaging should be performed at diagnosis. |
| **Congenital heart disease:** | Echocardiogram should be performed at diagnosis to screen for congenital heart defects |
| **Gastrointestinal:** | We recommend elective gastrostomy tube placement to support growth and limit pulmonary aspiration. An upper gastrointestinal study and renal ultrasound should be considered to screen for congenital anomalies. |
| **Speech and language:** | Speech and language therapy should be commenced at an early stage to maximise neurocognitive outcome. |
| **Motor development:** | Physical and occupational therapies should be commenced at an early stage to maximise neurocognitive outcome. |
| **Vision/Hearing:** | Yearly ophthalmology and audiology assessments are recommended. |
| **Endocrine:** | Infants and children should be monitored for hypoglycaemia and hypothyroidism as these are easily treatable. |
| **Behavioural abnormalities:** | Behavioural therapy (including Applied Behaviour Analysis) is beneficial in older affected children who have autism spectrum disorder. |
| **Sleep:** | Many children experience sleep dysregulation. Melatonin has been used successfully in some children. |
| **Education:** | An assessment of special educational needs should be carried out so that an individualized educational plan can be put in place at school. Some children have behavioural difficulties requiring additional support and one-to-one instruction. |

These guidelines are based on the most commonly identified features in individuals with *SNIP1*-related disorder. There is a wide range in variability of the clinical presentation and individual patients should have a personalized plan to reflect their own clinical features.

developmental regression. These simple interventions can reduce hospitalizations, alleviate suffering, and prolong life. Supportive therapies (occupational, speech, and physical therapy) should be commenced at an early stage to maximise neurocognitive outcome for patients with *SNIP1*-related disorder. Behavioural therapy (including Applied Behaviour Analysis) is beneficial in older affected children who have autism spectrum disorder. We advise that an EEG should be obtained at an early stage in infants with *SNIP1*-related disorder and suspected apneic episodes. Neuroimaging should be performed at diagnosis, and echocardiogram should be undertaken to screen for congenital heart defects. An upper gastrointestinal study, head ultrasound, and renal ultrasound should also be considered to screen for other congenital anomalies. Individuals should also be monitored for hypoglycemia and hypothyroidism as these are easily treatable. Yearly ophthalmology and audiology assessments are recommended.

While the pathomolecular outcomes associated with the *SNIP1* NM_024700.4:c.1097A>G, p.(Glu366Gly) substitution remain unclear, *SNIP1* null mouse models display embryonic lethality indicating that this Amish variant may be unlikely to result in complete loss of function [14]. While no other biallelic variants in *SNIP1* have conclusively been associated with genetic disease to date, Jacher and Innis (2018) reported a 17-year-old female with a de novo 2.3 Mb interstitial deletion at 1q34.3q34.2 that encompassed 43 genes, of which only two (*SNIP1* and *RSPO1*) were potentially linked with genetic disease genes at the time of publication [15]. This female had global developmental delay, mild intellectual disability, congenital defects, vocal cord paralysis, delayed bone age, and skeletal deformities. The authors concluded that haploinsufficiency of *SNIP1* was the most likely cause of the female's neurodevelopmental phenotype, although they were not able to confirm this nor rule out contributions of the other genes involved. In addition to NM_024700.4:c.1097A>G, p.(Glu366Gly), ClinVar lists six further "pathogenic" entries for *SNIP1*: five entries as part of a multiple gene deletion, and one *SNIP1* missense variant (NM_024700.4:c.331C>T; p.(Arg111Cys)) in an individual with Rolandic epilepsy, with low allele frequency (0.00018) and no homozygotes in gnomAD v2.1/ v3.1. Previous studies and online gene expression databases (GTEx: https://gtexportal.org/ home/) show that *SNIP1* mRNA expression is ubiquitous, consistent with the diverse multisystem phenotypical outcomes associated with variation in *SNIP1*, with particularly high expression in neurological, cardiac and skeletal systems [3,8]. SNIP1 has been identified as a regulator of several transcription factors, including Smads (transcription factors in the TGF-β pathway), NF-kB (a transcription factor in proinflammatory and immune signalling pathways) and c-MYC (a transcription factor in cell proliferation and growth-promoting pathways) as well as binding co-activators for transcription, including CREBBP and EP300 [3–5]. The p. (Glu366Gly) SNIP1 amino acid substitution is located in close proximity to the forkhead association (FHA) domain and within the C-terminal 30 amino acids, entailing a 'low complexity or intrinsically disordered region' (IDR). Thus while this region lacks a well-defined 3D structure for modelling analyses, it is strongly conserved across multiple species (Fig 1G) and is highly likely to represent a functionally important region of SNIP1 [16].

In TGF-β/BMP pathway signalling, ligand binding to the TGF-β receptor results in phosphorylation of Smad 2/3 which bind to Smad4, resulting in nuclear translocation of the Smad complex and recruitment of co-activators EP300 and CREBBP to induce the expression of target genes. The SNIP1 C-terminus interacts with Smads 1/2, while the N-terminus NLS binds both Smad4 and CREBBP/EP300 coactivators; overexpression of *SNIP1* inhibits CREBBP/ EP300 and the formation of Smad4 complexes, thereby suppressing transcription and inhibiting multiple gene responses to TGF-β signalling [3]. Disorders involving Smads and other components of the TGF-β pathway include connective tissue, craniofacial and skeletal disorders, several of which display overlapping clinical features with *SNIP1*-related disorder detailed here. This includes systemic aortopathies, including for example Loeys-Dietz syndrome

(LDS), characterized by congenital cardiac defects including bicuspid aortic valve, septal defects, patent ductus arteriosus, mitral valve regurgitation, aortic root dilatation, in addition to craniosynostosis, cleft palate, hypertelorism, micrognathia, talipes, and scoliosis [17]. Additionally, other conditions associated with disruption of the TGF-β pathway include the brachdactyly syndromes (e.g. type A2, B2 and C), which display the overlapping clinical feature of brachydactyly. SNIP1 has also been implicated in the inhibition of transcription factor NF-kB by competitive binding with the RelA subunit for binding to the coactivator EP300. The NF-kB family of transcription factors are regulators of a large number of genes involved in immune and inflammatory processes, cell growth and development. NF-kB is composed of dimeric complexes of five different subunits, RelA, RelB, RelC, NF-kB1 and NF-kB2, which mediate gene transcription by binding kB enhancer with the coactivator EP300 [18]. Disruption of NF-kB in several animal models results in severe developmental defects during embryogenesis including dorsal-ventral patterning, limb, liver, skin, lung, neural, notochord, muscle, skeletal, and hematopoietic defects (reviewed in Espín-Palazón et al 2016) [19], again mirroring some outcomes identified in *SNIP1*-related disorder. Additionally other studies identified SNIP1 downregulation in individuals with dilated cardiomyopathic hearts, and in aortic banding-induced mice hearts [20], shown to be mediated by NF-kB signalling. Remarkably, SNIP1 inhibition of this pathway reduced cardiac hypertrophy, while blocking the NF-kB pathway reduced the adverse effects of SNIP1 deficiency; these findings are notable as left ventricular noncompaction is also a feature seen in *SNIP1*-related disorder.

Zhang et al confirmed that SNIP1 forms a component of the human activated spliceosome (B$^{act}$ complex) [9], as part of the retention and splicing (RES) complex with Bud13 and RBMX2 [21], which plays a crucial role in splicing and retention of pre-mRNA [10]. Disruption of spliceosome components have been associated with several neurodevelopmental disorders involving craniofacial defects, for example Guion-Almeida type mandibulofacial dysostosis, Nager syndrome, and cerebrocostomandibular syndrome, which also display several overlapping features with *SNIP1*-related disorder including midface hypoplasia, cleft palate, developmental delay, intellectual disability, congenital cardiac defects and scoliosis [22]. This identifies a further possible mechanism of disease pathogenesis in *SNIP1*-related disorder. To gain insight into the aberrant molecular processes which may derive from the NM_024700.4:c.1097A>G, p.(Glu366Gly) *SNIP1* variant and underlie this condition, we undertook whole transcriptome studies of individuals with *SNIP1*-related disorder. These studies were inherently limited due to the modest number of samples available for study and through the obvious limitations of comparing gene transcriptions profiles in blood, with tissues in which clinical outcomes are primarily manifested (brain). However such studies are warranted in genes such as *SNIP1* with ubiquitous tissue expression profiles, where blood studies may provide important insight into the key molecular signalling pathways likely influenced by genetic variation. Notably of the 11 upregulated and 32 downregulated gene expression profiles with an associated clinical phenotype which were most notably altered, 24 had a previously established association with neurological disease. This includes five genes (*ROBO1*, *SOX5*, *CNTNAP2*, *PAFAH1B1* and *TSNARE1*) associated with seizures, the cardinal feature of *SNIP1*-related disorder. Of these *ROBO1* (roundabout guidance receptor 1), the gene identified to be most upregulated in affected individuals with *SNIP1*-related disorder, is a member of the neural cell adhesion molecule (NCAM) family involved in axonal migration. Notably biallelic sequence variants within *ROBO1* are associated with a neurodevelopmental disorder displaying many of the cardinal features seen in *SNIP1*-related disorder including neurodevelopmental delay, white matter abnormalities [23], epilepsy, autistic features [24], VSD and tetralogy of Fallot [25]. The expression of *CNTNAP2* encoding contactin-associated protein 2 (CASPR2), a member of the neurexin superfamily involved in neuron-glia interactions [26],

was also notably increased by *SNIP1* gene variation. Disruption of *CNTNAP2* is associated with autosomal recessive Pitt-Hopkins-like syndrome (also known as cortical dysplasia focal epilepsy syndrome; MIM 610042) characterized by neurodevelopmental delay, hypotonia, hyporeflexia, seizures of multiple types, neuronal migration abnormalities, autistic features, and attention deficits [27,28]. One further gene expression profile strongly upregulated includes *SYT1* (synaptotagmin 1), an integral membrane protein of synaptic vesicles which binds calcium to trigger neurotransmitter release at the synapse [29]. Variation within the *SYT1* gene has previously been associated with an autosomal dominant neurodevelopmental disorder (Baker-Gordon syndrome; OMIM 618218) characterized by neurodevelopmental delay, hypotonia, autistic features, non-specific white matter abnormalities, feeding problems, dysmorphic features, scoliosis, strabismus, nystagmus and self-injurious behaviour [30,31]. Further to these, the other 40 gene expression profiles most influenced by variation in *SNIP1* includes two other genes (*PAFAH1B1* and *TOR1AIP1*) associated with brain structural abnormalities, five (*PLXND1*, *TNS1*, *CAVIN4*, *EYA4* and *TOR1AIP1*) associated with cardiac defects including several conotruncal heart defects and cardiomyopathy, and two (*LINGO2* and *SOX5*) with dysmorphic craniofacial features (Tables A and B in S2 Table) overlapping features of *SNIP1*-related disorder. Further we investigated whether any up or down regulated genes were known interactors in SNIP1-regulated pathways, defining two downregulated genes (*BMPR1A* and *ACVR1C*) involved in the TGF-β pathway, two dysregulated genes (*BLNK* and *IL8*, up and downregulated respectively) which are components of the NF-kB pathway, and one upregulated gene (*BCL-7A*) within the c-MYC pathway [32]. Reactome analysis identified that the most highly dysregulated pathway was the TGF-β receptor signalling in epithelial to mesenchymal transition pathway, further supporting the critical role of SNIP1 within this pathway. Given this, it seems likely that the NM_024700.4:c.1097A>G, p.(Glu366Gly) SNIP1 substitution likely results in the clinical outcomes seen through the disrupted regulation of multiple molecular roles and signalling pathways important for human development, potentially defining candidate therapeutic targets for future research for *SNIP1*-related disorder. Taken together, these findings consolidate biallelic variants within the *SNIP1* gene as a cause of a complex neurodevelopmental disorder and provide important insight into the biological role and key molecular processes perturbed by pathogenic variation of SNIP1.

## Materials and methods

### Ethics statement

This study was carried out in accordance with institutional ethics review board-approved research protocols from the University of Arizona (10-0050-10) and Akron Children's Hospital (IRB 986876–3). All individuals (or their families) whose data is included in this study provided written informed consent and where applicable specific written consent for publication of photographs.

### Clinical and genetic studies

We identified a total of 51 individuals of Old Order Amish descent affected with *SNIP1*-related disorder through clinics who provide care for the Amish communities (The Community Health Clinic; Indiana, New Leaf Center; Ohio, Center for Special Children; Wisconsin, Clinic for Special Children; Pennsylvania). We conducted a historical review of clinical information and undertook physical examination and a parent survey for 35 of the affected children available for evaluation considering geographical restrictions. Blood/buccal samples were obtained for DNA extraction using standard techniques from affected individuals, their parents and unaffected siblings with informed consent. Single-nucleotide polymorphism (SNP) genotyping

was performed using HumanCytoSNP-12 (v2.1 beadchip array, Illumina). WES (BGI Seq) involved: Agilent Sureselect Whole Exome v6 targeting, read alignment (BWA-MEM (v0.7.17), mate-pairs fixed and duplicates removed (Picard v2.15.0), InDel realignment/base quality recalibration (GATK v3.7.0), single-nucleotide variant (SNV)/InDel detection (GATK HaplotypeCaller), annotation (Alamut v1.10) and read depth (GATK DepthOfCoverage). Copy number variants (CNVs) were detected using both ExomeDepth (https://cran.r-project.org/web/packages/ExomeDepth) and SavvyCNV [33]. Dideoxy sequencing was undertaken using standard techniques.

## Transcript analyses

Analysis was carried out for six affected individuals (3 male, 3 female) on whom samples were available between the ages of 2 and 28 years, and six sex-matched controls. Approximately 2.5 ml of peripheral blood was collected from each participant using PAXgene Blood RNA Tubes (IVD) (PreAnalytiX GmbH, Hombrechtikon, Switzerland) for RNA extraction according to manufacturer's instructions. Samples were assessed for RNA quality and quantity by Nano-drop spectrophotometry (NanoDrop, Wilmington, DE, USA). 100ng of total RNA was reverse transcribed using SuperScript VILO cDNA Synthesis Kit (ThermoFisher, Waltham, MA, USA); each sample was prepared in duplicate replicates. Gene cards were run on the Quant-Studio 12K Flex Real-Time PCR System (ThermoFisher, Waltham, MA, USA), using amplification conditions entailing: a single cycle of 50°C for 2 minutes, a single cycle of 94.5°C for 10 minutes followed by 40 cycles of 97°C for 30 seconds and 59.7°C for 1 minute. RNA-Seq reads were aligned to the human reference genome (NCBI GRCh38) with Bowtie2 (v 2.3.5.1) in paired-end mode using the 'local' and 'very-sensitive' presets [34]. Post-processing including conversion to BAM format and sorting were performed with samtools [35]. Reads mapping to known genes were quantified using ht-seq (v 0.6.1p1) [36] with a PHRED quality score cutoff of 10 using the 'intersection-strict' model and the un-stranded assignment mode. Differentially expressed genes were identified using edgeR (v 3.8.6) [37]. Briefly, lowly expressed genes were filtered using a counts per million (CPM) cutoff of $>2$ in $> = 10$ samples, counts were normalised using the trimmed mean of M-values and dispersion was estimated using built in methods. In order to calculate the power to detect significantly differentially expressed genes in our data we used the R package RnaSeqSampleSize [38]. The function 'est_power' was used to perform the power calculation given the sequencing characteristics of our dataset. These sequencing characteristics included; number of samples in each group (n = 6), minimum fold change for prognostic genes (rho = 1.85), total number of genes for testing (m = 12,694), number of expected DE genes (m1 = 187; from our differential expression analysis), false discovery rate (f = 0.01), dispersion of genes (phi0 = 0.1) and average read count for differential expression (lambda0 = 1791). Finally, differentially expressed genes were identified using the exact test and p-values were false discovery rate (FDR) corrected using the method of Benjamini and Hochberg [39]. Genes with an adjusted p-value $<0.05$ were deemed to be significantly differentially expressed. Differentially expressed (both up/downregulated) genes in affected individuals compared to controls with a *log(fold change)* $>1.5$ were used in gene set enrichment analysis [40] of pathway and gene ontology using Reactome [13] and pathways with an entities p value $<0.05$ were considered significantly overrepresented.

## Supporting information

**S1 Text. Patient EEG report findings for eight individuals affected with *SNIP1*-related disorder.**
(DOCX)

**S2 Text. ACMG Variant Classification Criteria.**
(DOCX)

**S1 Fig. Genome-wide SNP mapping in four affected individuals (X:30, X:31, XI:6 and XI:7) identified a single (1.65Mb) region of shared homozygosity, containing 17 genes.**
(DOCX)

**S2 Fig. Heatmap showing tissue expression levels of the top 50 genes with upregulated expression in individuals with *SNIP1*-related disorder.** Map created in GTex Portal, Broad Institute 2021.
(DOCX)

**S3 Fig. Heatmap showing tissue expression levels of the top 50 genes with downregulated expression in individuals with *SNIP1*-related disorder.** Map created in GTex Portal, Broad Institute 2021.
(DOCX)

**S1 Table. List of genes that are differentially expressed in individuals with *SNIP1*-related disorder.** Genes were identified by whole transcriptome sequencing studies of six affected individuals and sex-matched controls, with a false discovery rate (FDR) adjusted p-value of <0.05 and ranked by log fold change (logFC). **(A)** List of 75 significantly upregulated genes. **(B)** List of 109 significantly downregulated genes.
(XLSX)

**S2 Table. Genes with an identified clinical association that are up/downregulated in individuals with *SNIP1*-related disorder.** Tissues expressing each gene, including tissue of highest expression and known clinical associations are detailed. **(A)** List of 11 genes with an associated clinical phenotype that are upregulated (*log fold change (logFC)* >1.5) in individuals with *SNIP1*-related disorder. **(B)** List of 32 genes with an associated clinical phenotype that are downregulated (*log fold change (logFC)* <-1.5) in individuals with *SNIP1*-related disorder. GU; genitourinary, GTC; generalised tonic clonic seizures, SLE; systemic lupus erythematosus, SBH; subcortical band heterotopia, DSD; disorder of sexual development, CRC; colorectal cancer.
(XLSX)

**S3 Table. Differentially expressed (both up/downregulated) gene pathways from Reactome overrepresentation analysis of significantly up/downregulated genes in six affected individuals compared with sex-matched controls and a *log(fold change)* >1.5 were used in gene set enrichment analysis of pathway and gene ontology.** Pathways with an entities p value <0.05 were considered significantly dysregulated. **(A)** List of upregulated gene pathways. **(B)** List of downregulated gene pathways. **(C)** List of combined dysregulated gene pathways (both up/downregulated). #Entities found: the number of curated molecules that are common between the submitted data set and the named pathway. #Entities total: The total number of curated molecules within the pathway. Entities ratio: The proportion of Reactome pathway molecules represented by this pathway. Calculated as the ratio of entities from this pathway that are found in the submitted data set Vs. all entities within the pathway. Entities pvalue: The result of the statistical test for over-representation: the probability that the overlap between the query and the pathway has occurred by chance. Entities FDR: False discovery rate. Corrected over-representation probability. #Reactions found: The number of reactions in the pathway that are represented by at least one molecule in the submitted data set. #Reactions Total: The number of reactions in the pathway. Reactions ratio: The proportion of Reactome reactions

represented by this pathway. Calculated as the ratio of reactions from this pathway that are found within the submitted data set Vs. all Reactome reactions within the pathway. Submitted entities found: genes identified from the submitted list.
(XLSX)

## Acknowledgments

First and foremost we are grateful to the Amish families for their participation in this study and support of the Windows of Hope project. The authors thank the Anabaptist Variant Server collaborators; Regeneron Genetics Center, University of Maryland, Clinic for Special Children, Das Deutsche Clinic, National Institute of Mental Health Amish Program, for providing summary measures from genomic data.

## Author Contributions

**Conceptualization:** Zineb Ammous, Emma L. Baple, Andrew H. Crosby.

**Data curation:** Zineb Ammous, Lettie E. Rawlins, Hannah Jones, Joseph S. Leslie, Olivia Wenger, Ethan Scott, Jim Deline, Tom Herr, Rebecca Evans, Angela Scheid, Barry A. Chioza, Erik G. Puffenberger.

**Formal analysis:** Zineb Ammous, Lettie E. Rawlins, Ryan M. Ames, Lorna Harries.

**Funding acquisition:** Zineb Ammous, Lettie E. Rawlins, Ryan M. Ames, Emma L. Baple, Andrew H. Crosby.

**Investigation:** Zineb Ammous, Lettie E. Rawlins, Hannah Jones, Joseph S. Leslie, Olivia Wenger, Ethan Scott, Jim Deline, Tom Herr, Rebecca Evans, Angela Scheid, Barry A. Chioza, Ryan M. Ames, Harold E. Cross, Erik G. Puffenberger, Lorna Harries, Emma L. Baple, Andrew H. Crosby.

**Methodology:** Joseph S. Leslie, Lorna Harries, Emma L. Baple, Andrew H. Crosby.

**Project administration:** Emma L. Baple, Andrew H. Crosby.

**Software:** Ryan M. Ames.

**Supervision:** Zineb Ammous, Lorna Harries, Emma L. Baple, Andrew H. Crosby.

**Visualization:** Zineb Ammous, Emma L. Baple, Andrew H. Crosby.

**Writing – original draft:** Zineb Ammous, Lettie E. Rawlins, Emma L. Baple, Andrew H. Crosby.

**Writing – review & editing:** Zineb Ammous, Lettie E. Rawlins, Joanna Kennedy, Emma L. Baple, Andrew H. Crosby.

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
