## [Decision Letter · Decision Letter 0]

27 Jul 2021

Dear Dr Crosby,

Thank you very much for submitting your Research Article entitled 'SNIP1 alteration underlies a complex neurodevelopmental disorder' to PLOS Genetics.

The manuscript was fully evaluated at the editorial level and by independent peer reviewers. The reviewers appreciated the attention to an important topic but identified some concerns that we ask you address in a revised manuscript

We therefore ask you to modify the manuscript according to the review recommendations. Your revisions should address the specific points made by each reviewer.

[LINK]

Yours sincerely,

Gregory M. Cooper, PhD

Associate Editor

PLOS Genetics

Gregory Barsh

Editor-in-Chief

PLOS Genetics

Reviewer's Responses to Questions

**Comments to the Authors:**

Reviewer #1: Ammous, et al. describe a large cohort of Amish individuals with homozygous variation affecting one amino acid residue of SNIP1. While two publications have previously associated this gene with a complex NDD (with recognizable clinical features including seizures, craniofacial features, etc.), this paper adds detailed phenotypic data for many additional individuals. The authors also performed RNA-seq on a subset of individuals and suggest pathways involved in pathogenesis. The strengths of the manuscript include detailed phenotypic data for a large cohort, allowing creation of care recommendations. The study also provides a notable (although small) RNA-seq dataset that may aid in additional research opportunities.

My concerns are mostly regarding the novelty of the gene-disease relationship; I would suggest changing the title of the paper to reflect previous work that has been done in this area, and that this is an expansion of that work. Also, it is not clear if the 3 individuals from Puffenberger, et al. were included as part of the 35 here, although it seems like at least one is. Fig1E here looks identical to that of Puffenberger, et al.’s Figure 3B. De-identified images that were covered in the pdf may also be replicated. It would be nice to show data in that Figure from probands other than those that have been previously published, and generally distinguish what was previously done from what is new here.

Other detailed comments:

1. Either add the RefSeq NM and version, or remove the c./p. when just referencing the variant throughout the text. Do leave all the details on p.11, line 196.

2. change “consolidate” in abstract (further support? Solidify?)

3. Overall I would like clarification on the number of individuals and their relatedness. I was a bit confused by the sample numbers at the beginning of the discussion. Why weren’t all 49 individuals who had homozygous SNIP1 variation included? Due to limitations of clinical info? It may be useful to move this summary to the beginning of the results to clarify the population. I like that you mentioned there that these 35 are from 21 interrelated families (p.7, line 120), but maybe mention here a bit more on relatedness of individuals. Something like how many are not first-degree relatives/at least as distant as first cousins/two segregations/etc.

4. “affected with SNIP1-related disorder” (p.7, line 120)…does this mean based on phenotype/clinical features? Would “suspected SNIP1-related disorder” be more accurate?

5. Only a 60% probability of finding significantly differentially expressed genes in RNA-seq seems notably low. Can sample size be increased? (Authors do note that samples size is low.) Did all samples come from that one family in Figure 1F? Is there any possibility of using another individual from a different family?

6. Regarding ClinVar (p.16, line 290): There are other VUSs as well…can you comment on predictions for these (Near NLS, FHA, etc.? Any comment on their frequency in Amish populations? (And frequency of the R111C variant?)

7. Were any other variants of interest identified in WES that were outside of the chr1 region of interest?

8. Have you tried to identify others with variation of interest in SNIP1 through GeneMatcher?

9. It might be nice to mention this variant is rare in gnomAD and TopMed/Bravo, with no homozygous individuals. In gnomAD, 10/11 heterozygotes are in an Amish population.

Reviewer #2: The authors describe in detail clinical findings in 35 individuals from the Amish community with homozygous pathogenic SNIP1 variants (c.1097A>G p.(Glu366Gly)) and define the cardinal features of the disorder. They report also on gene transcript studies in affected individuals. The paper is enjoyable, robust and well written.

The authors claim that they ‘describe extensive genetic studies and clinical findings of a complex inherited neurodevelopmental disorder in 35 individuals associated with a SNIP1 c.1097A>G p.(Glu366Gly) alteration, present at high frequency in the Amish community’. They report that ‘together these data consolidate SNIP1 gene alteration as a cause of an autosomal recessive complex neurodevelopmental disorder and provide important insight into the molecular roles of SNIP1 which likely explain the cardinal clinical outcomes in affected individuals, defining potential therapeutic targets’.

This is not an original gene discovery paper as this gene is already an OMIM morbid gene, the gene is a possible gene in DDG2P and amber on UK PanelApp following work by Puffenberger et al. (2012) which identified exactly the same pathogenic variant identified in this study (c.1097A>G p.(Glu366Gly) and carried out functional work). However, it is an original thorough careful analysis of a large number of affected individuals which gives further very important evidence that biallelic pathogenic SNIP1 variants cause a distinctive disorder (SNIP1 associated syndrome). This will be extremely important for clinicians and scientists researching in the field

The authors claim to have defined therapeutic targets in the abstract, however this isn’t specifically referred to in the discussion (however they have carried out transcript analysis to identify perturbed pathways). This could be misread as meaning a specific molecule or receptor had been identified for targeting as opposed to large pathways which is what I think they are referring to. I think this could be reworded and referred to more clearly in the discussion

The authors have referenced the published literature appropriately and accurately; however, they could make it clearer that the pathogenic variant identified in this report has already been published.

Using ACMG criteria to classify the variant would make this paper better. This would take minimal time (less than one day)

It would have been helpful to give more information about the background of the Amish population and Old Order Amish as this is not given. It would be helpful to clinicians to highlight the spliceosomal aspects of this gene in the abstract so they can place it in context.

It isn’t totally clear how the authors identified the affected individuals. It would be helpful to state How exactly did you identify them? In clinic / examining them through the community? They refer at one point to a historical review of notes and parental survey. More clarity throughout would be helpful to the reader.

The use of ‘alteration’ in the title and paper is not the most accurate scientific/clinical word. Likewise, the title could be improved, and the authors could consider use of ‘SNIP1-associated syndrome’ which they use later in the paper.

With regards to epilepsy, were there any defining aspects of the individuals’ epilepsy. Any EEG abnormalities? Photosensitivity ? febrile seizures Are there any medications that any individuals responded well to? Information about treatment is very helpful to epileptologists.

The authors state several individuals required more than two antiepileptic drugs to achieve ‘reasonable’ seizure control. They don’t state how they defined reasonable seizure control. Do they mean: No seizures? Family reported happy with control? Doctor happy with control? No drop attacks?

The authors state: Individuals identified in late childhood/adolescence are often diagnosed with a mixed tone spastic quadriplegic cerebral palsy. Is this progressive? Why is this not picked up earlier?

The authors state: Other clinical features associated with SNIP1-related disorder include a variable spectrum of congenital cardiac defects (60% of individuals), Are the authors certain that all of these heart defects are all related to SNIP-1 and not due to other homozygous pathogenic variants in other genes? Are isolated heart defects more common in the Amish population than in the general population or not?

There are a small number of typographical errors. Figure 1 looks unfinished, parts A to D are missing. These appear to be clinical photos

**Have all data underlying the figures and results presented in the manuscript been provided?**

Reviewer #1: Yes

Reviewer #2: **No: **There is missing data from figure 1 (1a,b,c,d), I understand these are clinical images.

PLOS authors have the option to publish the peer review history of their article (what does this mean?). If published, this will include your full peer review and any attached files.

Reviewer #1: No

Reviewer #2: No

---

## [Editor Report · Decision Letter 1]

2 Sep 2021

Dear Dr Crosby,

We are pleased to inform you that your manuscript entitled "A biallelic SNIP1 Amish founder variant causes a recognizable neurodevelopmental disorder" has been editorially accepted for publication in PLOS Genetics. Congratulations!

Yours sincerely,

Gregory M. Cooper, PhD

Associate Editor

PLOS Genetics

Gregory Barsh

Editor-in-Chief

PLOS Genetics

Comments from the reviewers (if applicable):

**Data Deposition**

http://datadryad.org/submit?journalID=pgenetics&manu=PGENETICS-D-21-00722R1

**Press Queries**

---

## [Editor Report · Acceptance letter]

22 Sep 2021

PGENETICS-D-21-00722R1 

A biallelic *SNIP1* Amish founder variant causes a recognizable neurodevelopmental disorder 

Dear Dr Crosby, 

We are pleased to inform you that your manuscript entitled "A biallelic *SNIP1* Amish founder variant causes a recognizable neurodevelopmental disorder" has been formally accepted for publication in PLOS Genetics! Your manuscript is now with our production department and you will be notified of the publication date in due course.

With kind regards,

Andrea Szabo

PLOS Genetics

On behalf of:
